# Critical Appraisal of Leibovich 2018 and GRANT Models for Prediction of Cancer-Specific Survival in Non-Metastatic Chromophobe Renal Cell Carcinoma

**DOI:** 10.3390/cancers15072155

**Published:** 2023-04-05

**Authors:** Mattia Luca Piccinelli, Simone Morra, Stefano Tappero, Cristina Cano Garcia, Francesco Barletta, Reha-Baris Incesu, Lukas Scheipner, Andrea Baudo, Zhe Tian, Stefano Luzzago, Francesco Alessandro Mistretta, Matteo Ferro, Fred Saad, Shahrokh F. Shariat, Luca Carmignani, Sascha Ahyai, Derya Tilki, Alberto Briganti, Felix K. H. Chun, Carlo Terrone, Nicola Longo, Ottavio de Cobelli, Gennaro Musi, Pierre I. Karakiewicz

**Affiliations:** 1Cancer Prognostics and Health Outcomes Unit, Division of Urology, University of Montréal Health Center, Montréal, QC H2X 0A9, Canada; 2Department of Urology, IEO European Institute of Oncology, IRCCS, 20141 Milan, Italy; 3Department of Oncology and Haemato-Oncology, Università degli Studi di Milano, 20122 Milan, Italy; 4Department of Neurosciences, Science of Reproduction and Odontostomatology, University of Naples Federico II, 80131 Naples, Italy; 5Department of Urology, IRCCS Policlinico San Martino, 16132 Genova, Italy; 6Department of Surgical and Diagnostic Integrated Sciences (DISC), University of Genova, 16148 Genova, Italy; 7Department of Urology, University Hospital Frankfurt, Goethe University Frankfurt am Main, 39120 Frankfurt am Main, Germany; 8Division of Experimental Oncology, Unit of Urology, URI Urological Research Institute, IRCCS San Raffaele Scientific Institute, 20132 Milan, Italy; 9Martini-Klinik Prostate Cancer Center, University Hospital Hamburg-Eppendorf, 20246 Hamburg, Germany; 10Department of Urology, Medical University of Graz, 8036 Graz, Austria; 11Department of Urology, IRCCS Policlinico San Donato, 20097 Milan, Italy; 12Department of Urology, Comprehensive Cancer Center, Medical University of Vienna, 1090 Vienna, Austria; 13Department of Urology, Weill Cornell Medical College, New York, NY 10065, USA; 14Department of Urology, University of Texas Southwestern Medical Center, Dallas, TX 75390, USA; 15Hourani Center of Applied Scientific Research, Al-Ahliyya Amman University, Amman 19328, Jordan; 16Department of Urology, IRCCS Ospedale Galeazzi-Sant’Ambrogio, 20157 Milan, Italy; 17Department of Urology, University Hospital Hamburg-Eppendorf, 20246 Hamburg, Germany; 18Department of Urology, Koc University Hospital, 34010 Istanbul, Turkey

**Keywords:** cancer-specific mortality, chromophobe kidney cancer, prognostic model

## Abstract

**Simple Summary:**

To date, guideline-recommended prognostic models predicting cancer-control outcomes in chromophobe kidney cancer patients have never been validated in a large-scale contemporary North American cohort. We addressed this knowledge gap and performed a formal validation of Leibovich 2018 and GRade, Age, Nodes and Tumor (GRANT) prognostic models with cancer-specific survival as an outcome. Moreover, we proposed a novel nomogram for the prediction of the same outcome.

**Abstract:**

Within the Surveillance, Epidemiology, and End Results database (2000–2019), we identified 5522 unilateral surgically treated non-metastatic chromophobe kidney cancer (chRCC) patients. This population was randomly divided into development vs. external validation cohorts. In the development cohort, the original Leibovich 2018 and GRANT categories were applied to predict 5- and 10-year cancer-specific survival (CSS). Subsequently, a novel multivariable nomogram was developed. Accuracy, calibration and decision curve analyses (DCA) tested the Cox regression-based nomogram as well as the Leibovich 2018 and GRANT risk categories in the external validation cohort. The accuracy of the Leibovich 2018 and GRANT models was 0.65 and 0.64 at ten years, respectively. The novel prognostic nomogram had an accuracy of 0.78 at ten years. All models exhibited good calibration. In DCA, Leibovich 2018 outperformed the novel nomogram within selected ranges of threshold probabilities at ten years. Conversely, the novel nomogram outperformed Leibovich 2018 for other values of threshold probabilities. In summary, Leibovich 2018 and GRANT risk categories exhibited borderline low accuracy in predicting CSS in North American non-metastatic chRCC patients. Conversely, the novel nomogram exhibited higher accuracy. However, in DCA, all examined models exhibited limitations within specific threshold probability intervals. In consequence, all three examined models provide individual predictions that might be suboptimal and be affected by limitations determined by the natural history of chRCC, where few deaths occur within ten years from surgery. Further investigations regarding established and novel predictors of CSS and relying on large sample sizes with longer follow-up are needed to better stratify CSS in chRCC.

## 1. Introduction

Among the different forms of surgically treated kidney cancer, 25% of cases are non-clear cell renal cell carcinoma (non-ccRCC). Chromophobe RCC (chRCC) is the second most common non-ccRCC subtype [1]. The accurate prediction of cancer control outcomes in RCC patients is important for counselling, the planning of follow-up and the selection of appropriate adjuvant trial designs. Several prognostic models incorporating clinical and pathological RCC variables have been validated to predict recurrence and mortality after nephrectomy in chRCC [2,3,4,5,6,7,8,9,10,11,12,13,14]. Of these, two models are recommended by European guidelines to predict cancer control outcomes after nephrectomy in non-metastatic chRCC [15]: Leibovich 2018 [2,16] and GRade, Age, Nodes and Tumor (GRANT) [3,16,17,18,19,20]. However, their ability to predict cancer-specific survival (CSS) has only been validated in a mono-institutional European cohort of 127 chRCC patients [16]. We assessed the ability of these prognostic models to predict CSS in a population-based North American cohort relying on the 2000–2019 Surveillance, Epidemiology, and End Results (SEER) database [21,22]. Moreover, we developed and externally validated a novel prognostic nomogram for individual prediction of CSS in chRCC patients. Finally, we performed a head-to-head comparison of the novel nomogram with Leibovich 2018 and GRANT risk categories within the same external validation cohort. We hypothesized that both validated models (Leibovich 2018 and GRANT) would demonstrate a high degree of accuracy and correlation between predicted and observed CSS rates and that both would outperform random predictions in decision curve analyses (DCA) [23,24]. We also hypothesized that a nomogram based on a large and contemporary cohort could exhibit optimal performance characteristics.

## 2. Materials and Methods

### 2.1. Study Population and Variables Definition

Within the SEER database (2000–2019), we identified chRCC (International Classification of Disease for Oncology [ICD-O] site code C64.9; ICD-O histology code 8317/3 [25]) patients aged ≥ 18 years, treated with either radical or partial nephrectomy for unilateral [26] RCC. Moreover, only patients with complete data regarding age at diagnosis, T stage, N stage, grade, tumor size and sarcomatoid differentiation were included. Leibovich 2018 risk categories were defined according to the following criteria: sarcomatoid features (No vs. Yes), perinephric or renal sinus fat invasion (No vs. Yes) and N stage (N0-X vs. N1). GRANT risk categories were defined according to age at diagnosis (>60 vs. ≤60), T stage (T1-2-3a vs. T3b-c-4), N stage (N0-X vs. N1) and grade (G1-2 vs. G3-4). Five- and ten-year CSS (death from RCC) represented the endpoints of interest in the development, as well as the external validation cohort.

### 2.2. Statistical Analyses

The population was randomly and equally divided between development (50%) and external validation (50%) cohorts. Within the development cohort, we applied Leibovich 2018 and GRANT risk categories and quantified the regression coefficients for the prediction of cancer-specific mortality (CSM). Moreover, univariable Cox regression models (CRMs) tested time to CSM according to all available prognostic variables in the development cohort. Subsequently, of all the statistically significant variables in univariable CRMs, only the most informative variables relying on Akaike’s information criterion were included in the multivariable model and represented the novel nomogram [27].

In all of the subsequent steps, the developed novel nomogram and the Leibovich 2018 and GRANT risk categories were tested in the external validation cohort. First, the accuracy values of the prognostic models were quantified using Heagerty’s concordance index [28]. Second, 5- and 10-year CSS predictions were plotted against observed CSS in calibration plots according to each prognostic model. Finally, DCA was applied to quantify and compare the performance of the prognostic models relative to random predictions of CSS.

All statistical tests were two-sided, with the level of significance set at *p* < 0.05, and were performed with R Software Environment for Statistical Computing and Graphics (R version 4.1.3, R Foundation for Statical Computing, Vienna, Austria) [29].

## 3. Results

### 3.1. Descriptive Characteristics

We identified 5522 surgically treated non-metastatic chRCC patients with unilateral tumors diagnosed between 2000 and 2019. Of the overall population, 50% (*n* = 2761) were randomly selected and included in the development cohort. The remaining 50% (*n* = 2761) represented the external validation cohort.

In the development cohort, the median age at surgery was 59 (49–68, Table 1, Appendix A). The most frequent stages were T1 (62%), followed by T2 and T3 (18%), followed by T4 (2%). Lymph node involvement was observed in 1% of patients and only 1% of the population harbored sarcomatoid features. The median tumor size was 45 (30–75) mm (Appendix A). In this cohort, 85 vs. 13 vs. 2% of patients were classified as Leibovich 2018 Group 1 vs. Group 2 vs. Group 3, respectively, and 82 vs. 18% of patients were classified as belonging to the GRANT Favorable vs. Unfavorable risk category, respectively.

Observed CSS rates were 96 and 93% at five and ten years after surgery in the development cohort (Figure 1a).

According to Leibovich 2018 risk categories, 10-year CSS rates were 95 vs. 86 vs. 37% for Group 1 vs. Group 2 vs. Group 3, respectively, in the development cohort (Figure 1b). According to GRANT risk categories, 10-year CSS rates were 91 vs. 85% for the Favorable vs. Unfavorable risk category, respectively, in the development cohort (Figure 1c). The external validation cohort virtually perfectly mirrored the development cohort and was used to externally validate Leibovich 2018 and GRANT models as well as the new nomogram predicting 5- and 10-year CSS in non-metastatic chRCC patients.

### 3.2. Application of Leibovich 2018 and GRANT Risk Categories within the Development Cohort to Predict Cancer-Specific Survival

Regarding Leibovich 2018 risk categories, regression models predicting CSM reported a hazard ratio (HR) of 3.3 and 17.0 for Group 2 and Group 3, respectively, with Group 1 as the reference (Table 2).

Both variables achieved independent predictor status. Regarding GRANT risk categories, regression models predicting CSM reported an HR of 3.0 for the Unfavorable risk category with the Favorable risk category as the reference and achieved independent predictor status.

### 3.3. Development of a Novel Nomogram to Predict Cancer-Specific Survival in Chromophobe Kidney Cancer

In the development cohort, of all assessable predictors of CSS, seven demonstrated statistical significance (age at surgery, African American race/ethnicity, G3 grade, sarcomatoid features, tumor size, T stage, and N stage). The individual predicted accuracy values ranged from 0.51 to 0.70. After the application of Akaike’s information criterion rules, four variables (age at diagnosis, T stage, tumor size, N stage) remained in the final nomogram (Table 3, Figure 2).

In the external validation cohort (*n* = 2761), the application of the Leibovich 2018 risk categories’ regression coefficients for the prediction of CSS resulted in an accuracy of 0.68 and 0.65 when 5- and 10-year predictions were made, respectively (Table 2). Conversely, the application of the GRANT risk categories regression coefficients for the prediction of CSS resulted in an accuracy of 0.64 and 0.64 when 5- and 10-year predictions were made, respectively. In the same cohort, the newly developed nomogram resulted in an accuracy of 0.83 for the prediction of CSS at five years (Table 3) and an accuracy of 0.78 for the prediction of CSS at ten years. Calibration plots testing the agreement between predicted and observed CSS were virtually the same for all three examined models (Figure 3 and Figure 4).

Specifically, all three examined models exhibited minimal departures from ideal predictions, except for the Leibovich 2018 5-year calibration, where a more pronounced degree of overestimation in Group 3 was recorded. In DCA, all three models performed better than random predictions. However, all three models also exhibited limitations within specific ranges of threshold probabilities. In consequence, no individual model outperformed its competitors (Figure 5).

## 4. Discussion

Leibovich 2018 and GRANT prognostic models are guidelines recommended for the prediction of cancer control outcomes after surgical treatment for non-metastatic chRCC [15]. To date, the ability of these models to predict CSS five and ten years after surgery in chRCC patients has never been tested and compared in a contemporary population-based North American cohort. We addressed this knowledge gap and, additionally, developed and externally validated a novel prognostic nomogram for individual prediction of 5- and 10-year CSS in non-metastatic chRCC patients. We hypothesized that both validated models (Leibovich 2018 and GRANT) would demonstrate a high degree of accuracy and correlation between predicted and observed CSS rates and that both would outperform random predictions in DCA. We also hypothesized that a nomogram based on a large and contemporary cohort could exhibit optimal performance characteristics. Our results showed several important findings.

First, to the best of our knowledge, this is the first population-based North American analysis addressing 5- and 10-year CSS predictions using Leibovich 2018 or GRANT risk categories in chRCC patients. Its sample size is the largest reported and described for surgically treated non-metastatic chRCC patients (overall cohort: 5522; development cohort: 2761; external validation cohort: 2761). Sample size is particularly relevant when less frequent histological subtypes such as chRCC are examined. Moreover, large sample size is also important, when the CSS rate is elevated and few cancer mortality events occur. Indeed, chRCC distinguishes itself from clear-cell RCC (ccRCC) by substantially smaller CSM rates [15]. In consequence, despite the rarity of chRCC relative to ccRCC, larger sample sizes of patients at risk of CSM are required in analyses addressing chRCC. These considerations are particularly important when early-stage chRCC patients are included and are clearly operational in surgically treated non-metastatic patients. Finally, the current sample size represents the most contemporary (2000–2019) surgically treated non-metastatic chRCC population. It is of note that the original development cohorts where Leibovich 2018 (year of diagnosis: 1980–2010; *n* = 222) and GRANT (year of diagnosis: 1994–2006; *n* = 303) prognostic models were first defined were both smaller and more historical than the current study population. As a result, remarkable differences in the distribution of patients’ demographic, clinical and pathological characteristics were recorded, when the present cohorts and previous validation cohorts were compared. These differences are reflected, for example, in the distribution of patients across Leibovich risk categories (Group 1: 85 vs. 87%, Group 2: 13 vs. 7%, Group 3: 2 vs. 7% in the present cohort vs. Leibovich 2018 original cohort, respectively). These discrepancies further corroborate the need for testing and comparing the ability to predict CSS with Leibovich 2018 or GRANT risk categories in a large contemporary population-based North American chRCC cohort. This step is crucial prior to the clinical implementation of these models in North American surgically treated non-metastatic chRCC patients.

Second, we performed an external validation of Leibovich 2018 and GRANT risk categories predicting 5- and 10-year CSS rates in non-metastatic chRCC patients. Specifically, Leibovich 2018 and GRANT risk categories resulted in an accuracy of 0.68 and 0.64, respectively, for the prediction of CSS five years after surgery. Similarly, Leibovich 2018 and GRANT risk categories resulted in an accuracy of 0.65 and 0.64, respectively, for the prediction of CSS ten years after surgery. The risk categories exhibited very good calibration except for Leibovich 2018 5-year calibration, where a more pronounced degree of overestimation in Group 3 was recorded. Finally, Leibovich 2018 and GRANT risk categories outperformed random predictions in DCA. Taken together, these observations indicate borderline accuracy with very good calibration characteristics of both Leibovich 2018 and GRANT risk categories. DCA validated the use of these two models instead of random predictions. It should be noted that both models were initially devised to predict cancer recurrence; in consequence, it is possible that within a North American database reporting recurrence data, they could result in better accuracy and equally good calibration using recurrence-free survival as an outcome. This represents the main limitation of the current study. However, the National Cancer Database (NCDB) cannot be used to assess the accuracy of the model in predicting cancer-specific outcomes, since neither CSS nor cancer recurrence are available and overall mortality does not represent an adequate endpoint in surgically treated non-metastatic chRCC, since an important proportion of such patients would succumb to other cause mortality. Finally, it is unlikely that institutional databases would provide sufficiently large sample sizes to accomplish analyses addressing recurrence and CSS in a contemporary cohort. Rosiello et al. [16] previously tested the accuracy of Leibovich 2018 and GRANT in predicting 5-year CSS in chRCC, relying on a single institution European chRCC cohort (*n* = 127; year of diagnosis: 1987–2019). In their experience, accuracy values for both Leibovich 2018 and GRANT models were higher (Leibovich 2018: 0.88; GRANT: 0.80). However, their analysis was based on substantially smaller sample sizes and more historical observations in addition to selectively focusing on European patients.

Third, the comparison of the newly developed and externally validated nomogram relative to the external validation of Leibovich 2018 and GRANT risk categories revealed several noteworthy points. Specifically, the novel nomogram resulted in an accuracy of 0.83 for the prediction of CSS at five years after surgery (Leibovich 2018: 0.68, GRANT: 0.64) and an accuracy of 0.74 for the prediction of CSS at ten years after surgery (Leibovich 2018: 0.65, GRANT: 0.64) in an independent external validation cohort. The calibration of the novel nomogram was similar to GRANT risk categories and somewhat better than Leibovich 2018 in 5-year CSS prediction. However, in DCA, no clear winner could be identified. Specifically, within select ranges of threshold probabilities, Leibovich 2018 risk categories outperformed the novel nomogram. Conversely, the novel nomogram outperformed Leibovich 2018 in the remaining values of threshold probabilities. Accuracy, calibration and net benefit in DCA represent three essential ingredients that a novel model should exhibit in an independent external validation cohort [30].

Taken together, all models demonstrated important methodological limitations related to their performance. Even though the novel nomogram exhibited higher accuracy, it was outperformed in selected ranges of threshold probabilities by Leibovich 2018 risk categories when DCA was used as a benchmark. In consequence, no model can be clearly recommended above its competitors. Moreover, it should be noted that all models have important limitations that possibly question their applicability and usefulness in clinical practice. These limitations, which apply to all three examined models, are related to the natural history of surgically treated non-metastatic chRCC, where few deaths occur relative to ccRCC [11]. Moreover, the time to RCC-specific mortality is invariably much longer in chRCC than in ccRCC. In consequence, all prognostic models predicting cancer-control endpoints require very large chRCC cohorts, with very long follow-up. Such requirements may render cohorts with sufficiently mature data too outdated when model predictions based on their observations are generated, tested and reported. In consequence, the use of observational data that are stratified according to patient and tumor characteristics may prove more useful than the input of various prognostic models.

Despite its novelty, our study is not devoid of limitations. First, the SEER is a retrospective database with the potential for selection biases. However, observational databases such as SEER or NCDB represent ideal large-scale databases to study less frequent primaries, especially when rates of mortality are low. Second, no central review regarding pathological stage and histological subtype was applied within the SEER database. Last but not least, as previously discussed, no information assessing time to recurrence prevented us evaluate recurrence-free survival outcomes in addition to CSS.

## 5. Conclusions

Leibovich 2018 and GRANT risk categories exhibited borderline low accuracy in predicting CSS in North American non-metastatic chRCC patients. Conversely, the novel nomogram exhibited higher accuracy. However, in DCA, all of the examined models exhibited limitations within specific threshold probability intervals. In consequence, all three examined models provide individual predictions that might be suboptimal and affected by limitations determined by the natural history of chRCC, where few deaths occur within ten years after surgery. Further investigations regarding established and novel predictors of CSS and relying on large sample sizes with longer follow-up are needed to better stratify CSS in chRCC.

## Figures and Tables

**Figure 1 cancers-15-02155-f001:**
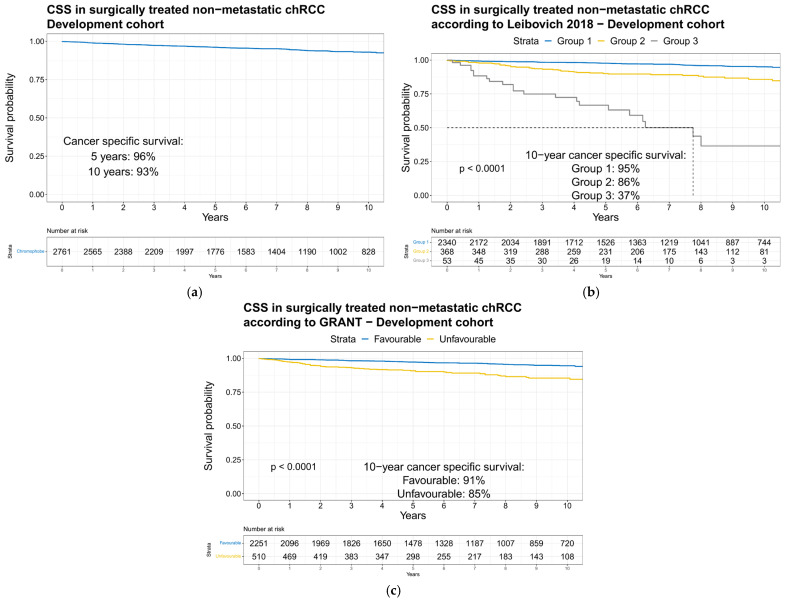
Development cohort: Kaplan–Meier plots with log-rank test depicting cancer-specific survival over ten years in patients with unilateral surgically treated non-metastatic chromophobe renal carcinoma diagnosed in 2000–2019 Surveillance, Epidemiology, and End Results database. (**a**) Overall; (**b**) according to Leibovich 2018 risk categories; (**c**) according to GRANT risk categories.

**Figure 2 cancers-15-02155-f002:**
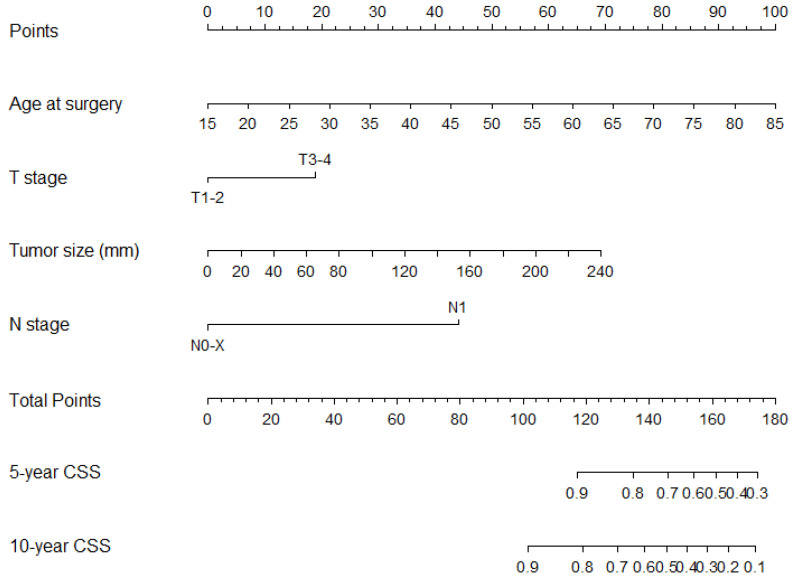
Nomogram predicting cancer-specific survival in surgically treated non-metastatic chromophobe renal cell carcinoma at five and ten years after surgery.

**Figure 3 cancers-15-02155-f003:**
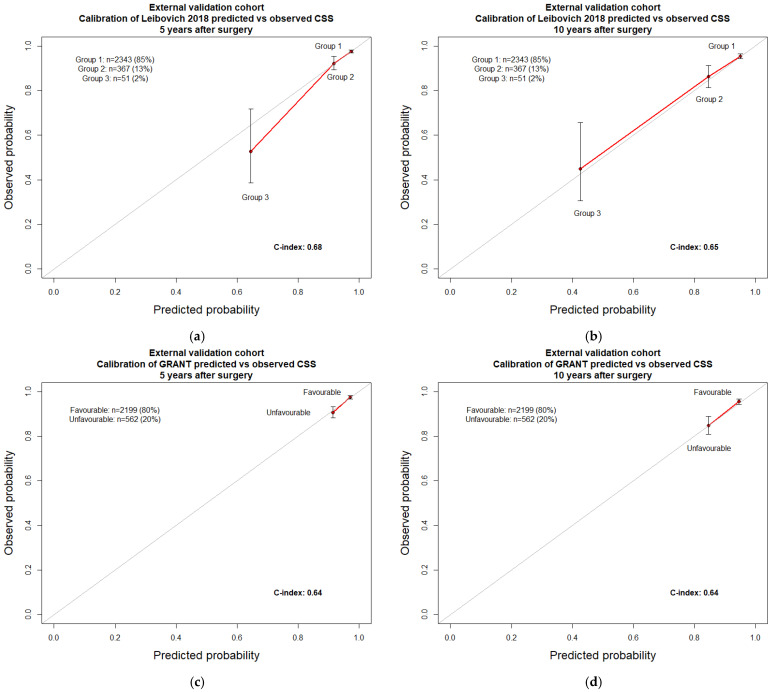
External validation cohort: Leibovich 2018 and GRANT prognostic models calibration plots. CSS: cancer-specific survival. (**a**) Cancer-specific survival at five years according to Leibovich 2018; (**b**) cancer-specific survival at ten years according to Leibovich 2018; (**c**) cancer-specific survival at five years according to GRANT; (**d**) cancer-specific survival at ten years according to GRANT.

**Figure 4 cancers-15-02155-f004:**
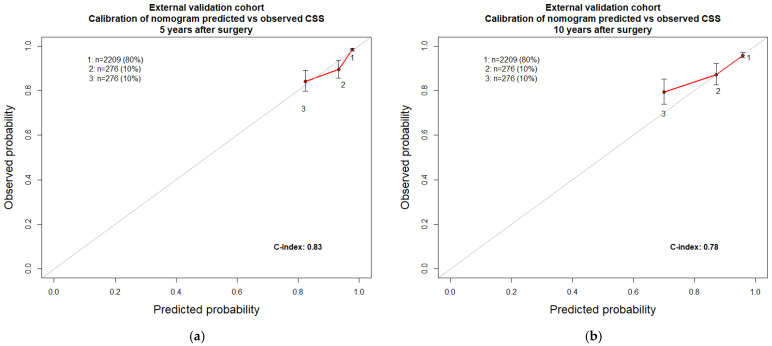
External validation cohort: nomogram calibration plots. CSS: cancer-specific survival. (**a**) Cancer-specific survival at five years; (**b**) cancer-specific survival at ten years.

**Figure 5 cancers-15-02155-f005:**
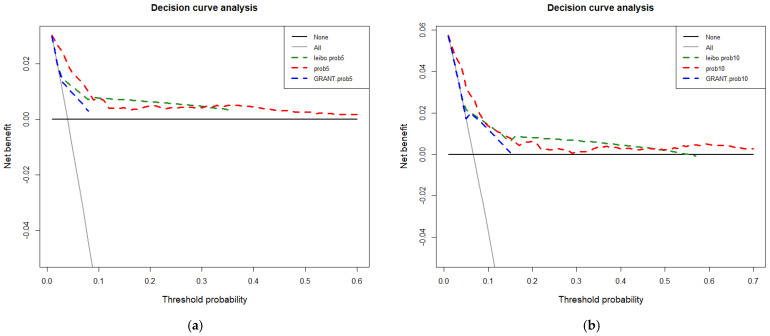
External validation cohort: decision curve analyses quantifying and comparing the performance of the prognostic models relative to random predictions of cancer-specific mortality: (**a**) 5-year survival; (**b**) 10-year survival.

**Table 1 cancers-15-02155-t001:** Descriptive characteristics of patients diagnosed with surgically treated unilateral non-metastatic chromophobe renal carcinoma between 2000 and 2019 in the Surveillance, Epidemiology, and End Results database. The overall cohort was randomly divided into a development (50%) and an external validation (50%) cohort. chRCC: chromophobe renal cell carcinoma.

Surgically Treated Non-Metastatic chRCC *n* = 5522	Development *n* = 2761	External Validation *n* = 2761
Age at surgery (years)		
Median (IQR)	59 (49–68)	60 (50–70)
18–35	140 (5%)	152 (6%)
36–50	613 (22%)	545 (20%)
51–70	1451 (53%)	1426 (51%)
≥71	557 (20%)	638 (23%)
Sex		
Male	1580 (57%)	1521 (55%)
Race/ethnicity		
Caucasian	1871 (68%)	1838 (67%)
African American	341 (12%)	379 (14%)
Hispanic	376 (14%)	369 (13%)
Asian or Pacific Islander	141 (5%)	143 (5%)
Other	32 (1%)	32 (1%)
Treatment		
Radical nephrectomy	1772 (64%)	1731 (63%)
Partial nephrectomy	989 (36%)	1030 (37%)
Grade		
G1	231 (8%)	237 (9%)
G2	1547 (56%)	1486 (54%)
G3	828 (30%)	864 (31%)
G4	155 (6%)	174 (6%)
Sarcomatoid features	31 (1%)	23 (1%)
T stage		
T1	1722 (62%)	1769 (64%)
T2	491 (18%)	476 (17%)
T3	503 (18%)	471 (17%)
T4	45 (2%)	45 (2%)
Size (mm)	45 (30–75)	45 (30–70)
Median (IQR)		
N stage		
N0-X	2737 (99%)	2728 (99%)
N1	24 (1%)	33 (1%)
Leibovich 2018 risk categories		
Group 1	2340 (85%)	2343 (85%)
Group 2	368 (13%)	367 (13%)
Group 3	53 (2%)	51 (2%)
GRANT risk categories		
Favourable	2251 (82%)	2199 (80%)
Unfavourable	510 (18%)	562 (20%)

**Table 2 cancers-15-02155-t002:** Separate univariable Cox regression models predicting cancer-specific mortality according to Leibovich 2018 risk categories and GRANT risk categories. For each model, c-indexes addressing the concordance between predicted and observed survival rates at five and ten years are provided. All patients were diagnosed with surgically treated unilateral non-metastatic chromophobe renal cell carcinoma between 2000 and 2019 in the Surveillance, Epidemiology, and End Results database.

Models Tested	Hazard Ratio	95% CI	*p*-Value	External Validation	
				5-Year c-Index	10-Year c-Index
**Leibovich 2018 risk categories**				0.68	0.65
Group 1	Ref				
Group 2	3.3	(2.3–4.7)	**<0.001**		
Group 3	17.0	(10.6–27.5)	**<0.001**		
**GRANT risk categories**				0.64	0.64
Favourable	Ref				
Unfavourable	3.0	(2.2–4.2)	**<0.001**		

Bold values indicate statistical significance *p* < 0.05.

**Table 3 cancers-15-02155-t003:** Univariable and multivariable Cox regression models predicting cancer-specific mortality. For each model, c-indexes addressing the concordance between predicted and observed survival rates at five and ten years are provided. All patients were diagnosed with unilateral surgically treated non-metastatic chromophobe renal cell carcinoma between 2000 and 2019 in the Surveillance, Epidemiology, and End Results database.

	Univariable					Multivariable				
Variables Tested	HR	95% CI	*p*-Value	Internal Validation 5-Year c-Index	Internal Validation 10-Year c-Index	HR	95% CI	*p*-Value	External Validation 5-Year c-Index	External Validation 5-Year c-Index
**Age at surgery (years)**	1.05	(1.04–1.06)	**<0.001**	0.65	0.69	1.06	(1.05–1.08)	**<0.001**	0.83	0.78
T stage				0.65	0.62					
T1-2	Ref					Ref				
T3-4	3.4	(2.5–4.6)	**<0.001**			2.3	(1.6–3.1)	**<0.001**		
**Tumour size (mm)**	1.01	(1.01–1.02)	**<0.001**	0.70	0.64					
**N stage**				0.54	0.54					
N0-X	Ref					Ref				
N1	10.8	(5.9–20.0)	<0.001			6.6	(3.4–12.6)	**<0.001**		
**Sex**				0.52	0.51					
Male	Ref									
Female	0.8	(0.6–1.1)	**0.2**							
**Race/ethnicity**				0.51	0.55					
Caucasian	Ref									
African American	1.6	(1.1–2.4)	0.02							
Hispanic	0.9	(0.6–1.5)	0.7							
Asian or Pacific Islander	0.96	(0.5–2.1)	0.9							
**Grade**				0.61	0.59					
G1	Ref									
G2	0.7	(0.4–1.3)	0.3							
G3	1.0	(0.5–1.8)	0.99							
G4	3.0	(1.5–5.9)	**0.001**							
**Sarcomatoid features**				0.54	0.56					
No	Ref									
Yes	14.0	(7.5–26.1)	<0.001							

Bold values indicate statistical significance *p* < 0.05.3.4. Accuracy, Calibration and Decision Curve Analyses in the External Validation Cohort.

## Data Availability

The data presented in this study are available in this article.

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
