# Peer review of "Critical Appraisal of Leibovich 2018 and GRANT Models for Prediction of Cancer-Specific Survival in Non-Metastatic Chromophobe Renal Cell Carcinoma"

_cancers, 2023, doi:10.3390/cancers15072155_

Round 1

Reviewer 1 Report

In this study, the author tried to improve and verify the previously published prognostic model of chRCC based on the SEER database, and proposed a new nomogram. The whole manuscript has a clear idea, and the improved model has higher accuracy, but there are still some deficiencies that need to be clearly explained and revised.

1.The incidence of chRCC is very low in people under the age of 25, and the peak age of onset is about 60 years old, so I wonder why the author set the age of the included patients to ≥18 years old, and the authors should show the average age and distribution in different age groups.

2.In Table 3, the authors should not only show the five significant predictors, the HR and P values of other factors should also be included in the text.

3. What are the regression coefficients for the five predictors? And the C-index of univariable  Cox regression models does not seem to be very satisfactory.

4. When constructing the nomogram, why are T stages divided into T1-2 and T3-4?

5.A calibration curve should be drawn to correct the predicted and actual values of the nomogram.

6. The font in the figures in the manuscript is too small.

Author Response

Response to Reviewer 1 Comments

In this study, the author tried to improve and verify the previously published prognostic model of chRCC based on the SEER database, and proposed a new nomogram. The whole manuscript has a clear idea, and the improved model has higher accuracy, but there are still some deficiencies that need to be clearly explained and revised.

We thank the reviewer for the comment.

Point 1: The incidence of chRCC is very low in people under the age of 25, and the peak age of onset is about 60 years old, so I wonder why the author set the age of the included patients to ≥18 years old, and the authors should show the average age and distribution in different age groups.

Response 1: We thank the reviewer for the very insightful comment. Indeed, chRCC is rare under 25 years of age. The median age at surgery was 59 (49-68) in the development cohort and 60 (50-70) in the external validation cohort as reported in Table 1. Specifically, 19 (0.7%) and 23 (0.8%) patients were ≤25 years old at surgery. We decided to include in our cohort patients aged ≤25 since both Leibovich 2018 and GRANT risk categories apply to patients aged ≥18. In consequence, the inclusion of patients aged ≥18 in the development and external validation of the novel nomogram was necessary to perform a head-to-head comparison between the guideline-recommended risk categories (Leibovich 2018 and GRANT) and the novel nomogram in the exact same cohorts.
To better describe age distribution in our cohorts we added in Table 1 the distribution of patients across four age groups: 18-35 (5% in the development cohort and 6% in the external validation cohort), 36-50 (22% in development cohort and 20% in external validation cohort), 51-70 (53% in development cohort and 51% in external validation cohort) and ≥71 (20% in development cohort and 23% in external validation cohort). We further depicted age distribution in with a density plot in Supplementary figure 1. We hope that the Reviewer will share our perspectives and find the above explanations satisfactory.

Point 2: In Table 3, the authors should not only show the five significant predictors, the HR and P values of other factors should also be included in the text.

Response 2: We thank the reviewer for the comment. We added HRs and p-values for all available risk factors in Table 3. We selected the most parsimonious model with the highest accuracy after permutations and Akaike’s information criterion definition. We initially excluded univariable HR and c-index of all other risk factors for the sake of synthesis. We hope that the Reviewer will share our perspectives and find the above explanations satisfactory.

Point 3: What are the regression coefficients for the five predictors? And the C-index of univariable  Cox regression models does not seem to be very satisfactory.

Response 3: We thank the reviewer for the comment. We selected four predictors in the final multivariable model depicted by the novel nomogram: age at surgery, T stage, tumor size and N stage. The multivariable Cox regression model coefficients were: 0.06 for age at surgery, 0.8 for T stage T3-4, 0.01 for tumor size and 1.9 for N stage N1. As stated by the reviewer, the accuracy of univariable models testing the association of risk factors with CSM is fair but not ideal. Usual staging, based on a separate assessment of T stage, N stage and tumor size, does not provide satisfactory accuracy in identifying patients that will more likely harbor a CSM event. As a result, multivariable prognostic models including simultaneously multiple risk factors and generating a more accurate prediction of CSS are clearly needed in chRCC. Focusing on the c-index, the novel multivariable prognostic model outperformed both usual staging factors (for example, T and N stage) and Leibovich 2018 and GRANT risk categories. We hope that the Reviewer will share our perspectives and find the above explanations satisfactory.

Point 4: When constructing the nomogram, why are T stages divided into T1-2 and T3-4?

Response 4: We thank the reviewer for the very insightful comment. chRCC patients were most frequently staged as T1 (development: 62%, validation: 64%), followed by T2 (development: 18%, validation: 17%) and T3 (development: 18%, validation: 17%), followed by T4 (development: 2%, validation: 2%), in that order. Additionally, 10-year CSS rates according to T stage in the development cohort were 96% for T1, 93% for T2, 85% for T3 and 83% for T4 patients. Finally, in univariable Cox regression model testing the association between T stage and CSM HRs were 2.1, 4.1 and 4.8 for T2, T3 and T4 patients, respectively. Based on these observations we decided to re-define T stage as T1-2 vs T3-4 for two reasons:

  • T1 and T2 tumors exhibited similar 10-year CSS rates. Conversely, in univariable Cox regression models T2 patients exhibited a two-fold higher risk of CSM. However, T1 and T2 definitions are based on tumor size. We included continuously coded tumor size (mm) in the novel prognostic model. A continuously coded variable invariably provides a more detailed risk stratification, when compared to cut-offs (as, for example, 4, 7 and 10 cm size cut-offs defined by TNM). We further explored the association between tumor size and CSM in chRCC with restricted cubic spline method (2, 3, 4 and 5 knots) in order to find accurate and clinically meaningful novel size cut-offs. However, we observed a linear increase of CSM risk corresponding to increasing tumor size with no identifiable statistically or clinically meaningful size cut-offs.
  • T3 and T4 tumors exhibited similar 10-year CSS rates and comparable HRs in the univariable Cox regression model. Moreover, T4 chRCC represents an extremely rare entity in our population and institutional cohorts analyzed by Leibovich et al. (<1%) and Rosiello et al. (0%) and reported as references. Based on these assumptions, we decided to group T3 and T4 tumors as a single level in order to maximize the generalizability of risk increase determined by locally advanced disease and not to rely on extremely infrequent T4 stage alone to stratify the risk of CSM on exceptional conditions.

Taken together, we stratified the T stage as T1-2 to not overlap the CSM risk contribution of tumor size, already provided by continuously coded tumor size (mm), with T1 and T2 stages coded separately. Conversely, we stratified the T stage as T3-4 to include the risk contribution of locally advanced disease. We grouped T3 with T4 since T4 chRCC patients are extremely rare and only a negligible amount of patients would benefit from the specific risk assessment of T4 chRCC. We hope that the Reviewer will share our perspectives and find the above explanations satisfactory.

Point 5: A calibration curve should be drawn to correct the predicted and actual values of the nomogram.

Response 5: We thank the reviewer for the comment. Quantile calibration curves for the novel nomogram testing the agreement between predicted and observed CSS at five and ten years after surgery are reported in Figure 4 and discussed in the third point of discussion.

Point 6: The font in the figures in the manuscript is too small.

Response 6: We thank the reviewer for the comment. We enlarged the font in the figures.

Reviewer 2 Report

The manuscript entitled "Critical appraisal of Leibovich 2018 and GRANT models for prediction of cancer-specific survival in non-metastatic chromophobe renal cell carcinoma” by Piccinelli is very poor and not suitable for publication in Cancer Journal. However, author need to address the following major concern before re-submission for consideration.

Comments to author

Introduction should contain current status, conventional strategies and their limitation and how your work overcoming those limitations. SO The Induction need to be more convincing the novelty of the work.

The materials and method section need to be divided into subsections.

Figure 1, quality is poor. Text is not convenient for reader.

Figure 2 also poor in quality

Conclusion missing the major outcome of the study

Author Response

Response to Reviewer 2 Comments

The manuscript entitled "Critical appraisal of Leibovich 2018 and GRANT models for prediction of cancer-specific survival in non-metastatic chromophobe renal cell carcinoma” by Piccinelli is very poor and not suitable for publication in Cancer Journal. However, author need to address the following major concern before re-submission for consideration.

We thank the reviewer for the comment and we are sorry for the negative opinion regarding our work. We hope that the following modifications will improve the quality of our work and satisfy the reviewer.

Point 1: Introduction should contain current status, conventional strategies and their limitation and how your work overcoming those limitations. SO The Induction need to be more convincing the novelty of the work.

Response 1: We thank the reviewer for the comment. We modified lines 64-69 of the introduction as follows to better explain the limitation of the current prognostic models validated for the prediction of cancer control outcomes in chRCC.

Introduction, lines 67-74:
“Of these, two models are recommended by European guidelines to predict cancer control outcomes after nephrectomy in non-metastatic chRCC [11]: Leibovich 2018 [2,12] and GRade, Age, Nodes and Tumor (GRANT) [3,12,13]. However, their ability to predict cancer-specific survival (CSS) has been only validated in a mono-institutional European cohort of 127 chRCC patients [12]. We assessed the ability of these prognostic models to predict CSS in a population-based North American cohort relying on 2000-2019 Surveillance, Epidemiology, and End Results (SEER) database [14].”

We hope that the Reviewer will share our perspectives and find the above explanations satisfactory.

Point 2: The materials and method section need to be divided into subsections.

Response 2: We thank the reviewer for the comment. We divided the Material and methods section into “2.1. Study population and variables definition” and “2.2. Statistical analyses” subsections.

Point 3: Figure 1, quality is poor. Text is not convenient for reader.

Response 3: We thank the reviewer for the comment. We improved the Figure quality and enlarged the font.

Point 4: Figure 2 also poor in quality.

Response 4: We thank the reviewer for the comment. We improved the Figure quality.

Point 5: Conclusion missing the major outcome of the study.

Response 5: We thank the reviewer for the comment. We modified the conclusion as follows:

Conclusions, lines 330-338:
“Leibovich 2018 and GRANT risk categories exhibited borderline low accuracy in predicting CSS in North American non-metastatic chRCC patients. Conversely, the novel nomogram exhibited higher accuracy. However, in DCA all examined models exhibited limitations within specific threshold probability intervals. In consequence, all three examined models provide individual predictions that might be suboptimal and affected by limitations determined by the natural history of chRCC, where few deaths occur within ten years from surgery. Further investigations regarding established and novel predictors of CSS and relying on large sample sizes with longer follow-up are needed to better stratify CSS in chRCC.”

We hope that the Reviewer will share our perspectives and find the above explanations satisfactory.

Reviewer 3 Report

Critical appraisal of Leibovich 2018 and GRANT models for 2 prediction of cancer-specific survival in non-metastatic chromo-3 phobe renal cell carcinoma

Manuscript ID cancers-2261039

Comments to author:

The manuscript is interesting but not written with somewhat technical depth, results, and analysis. However the following issues need to resolve before publishing in this journal:

1.      In abstract “The novel prognostic nomogram was 0.78 accurate at ten years.” But how? There is no evidence on those writing.   

2.      Some acronym is not clearly discussed before use.

3.      In introduction, the background study is very poor. Need to improve it clearly. Authors should care about it.

4.      In Fig. 1, the graphical representation of the study is very poor. Authors should take extra care about it. Especially Fig. 1A, Fig. 1B and Fig. 1C levels are not visible.

5.      Overall, figure’s representations in the whole manuscript are not in acceptable level. Please take care about it.

6.      My suggestion is that Table 1 could be used as a supplementary file..  

7.      Authors need to show the correlation between different features for the applied datasets. This analysis has measured how strongly one feature implies the other.

8.      Authors should focus the density plots of the applied datasets. The density plot shows the smooth distribution of the points along the numeric axis. The peaks of the density plot are at the locations where there is the highest concentration of points.

9.      The discussion part of this manuscript is not concise written for the new readers. Please improve the discussion part of the manuscript to validate your findings.

10.  Add Comparison of the performance of the proposed model with the existing models for better understanding.

11.  Please revise all the grammatical errors in this manuscript carefully.

Author Response

Response to Reviewer 3 Comments

The manuscript is interesting but not written with somewhat technical depth, results, and analysis. However the following issues need to resolve before publishing in this journal:

We thank the reviewer for the comment. We hope that the following modifications will improve the quality of our work and satisfy the reviewer.

Point 1: In abstract “The novel prognostic nomogram was 0.78 accurate at ten years.” But how? There is no evidence on those writing.

Response 1: We thank the reviewer for the comment. As stated in previous sentences, the accuracy of the novel nomogram was computed at five and ten years after surgery in the external validation cohort. Specifically, as stated in the Materials and methods section, the accuracy values of the prognostic models were quantified using Heagerty's concordance index (c-index). We hope that the Reviewer will share our perspectives and find the above explanations satisfactory.

Point 2: Some acronym is not clearly discussed before use.

Response 2: We checked and spelled out all acronyms as they are first written in the text.

Point 3: In introduction, the background study is very poor. Need to improve it clearly. Authors should care about it.

Response 3: We thank the reviewer for the comment. We modified lines 64-69 of the introduction as follows to better explain the limitation of the current prognostic models validated for the prediction of cancer control outcomes in chRCC.

Introduction, lines 67-74:

“Of these, two models are recommended by European guidelines to predict cancer control outcomes after nephrectomy in non-metastatic chRCC [11]: Leibovich 2018 [2,12] and GRade, Age, Nodes and Tumor (GRANT) [3,12,13]. However, their ability to predict cancer-specific survival (CSS) has been only validated in a mono-institutional European cohort of 127 chRCC patients [12]. We assessed the ability of these prognostic models to predict CSS in a population-based North American cohort relying on 2000-2019 Surveillance, Epidemiology, and End Results (SEER) database [14].”

We hope that the Reviewer will share our perspectives and find the above explanations satisfactory.

Point 4: In Fig. 1, the graphical representation of the study is very poor. Authors should take extra care about it. Especially Fig. 1A, Fig. 1B and Fig. 1C levels are not visible.

Response 4: We improved the figures' quality and enlarged the font.

Point 5: Overall, figure’s representations in the whole manuscript are not in acceptable level. Please take care about it.

Response 5: We improved the figures' quality and enlarged the font.

Point 6: My suggestion is that Table 1 could be used as a supplementary file.

Response 6: We thank the reviewer for the suggestion. However, we think that Table 1 is crucial to better understanding demographic and pathologic characteristics distribution in chRCC patients and highlighting differences between the cohorts. Moreover, since our cohort represents one of the largest reported to date, these distributions could be of value, when chRCC in North America is analyzed.

Point 7: Authors need to show the correlation between different features for the applied datasets. This analysis has measured how strongly one feature implies the other.

Response 7: We thank the reviewer for the very insightful comment. We tested for interactions between variables included in the novel nomogram in a multivariable fashion. No significant interaction between age at surgery and T stage (p=0.2), age at surgery and tumor size (p=0.95), age at surgery and N stage (p=0.1), T stage and tumor size (p=0.6), T stage and N stage (p=0.9), tumor size and N stage (p=0.07) were observed.

Point 8: Authors should focus the density plots of the applied datasets. The density plot shows the smooth distribution of the points along the numeric axis. The peaks of the density plot are at the locations where there is the highest concentration of points.

Response 8: We thank the reviewer for the suggestion. We added to the supplementary materials the density plots of age at surgery and tumor size. Age at surgery was characterized by a normal distribution with a peak at 62 years after surgery. Conversely, tumor size distribution appears to be right skewed with a peak at 45 mm.

Point 9: The discussion part of this manuscript is not concise written for the new readers. Please improve the discussion part of the manuscript to validate your findings.

Response 9: We thank the reviewer for the comment. We wrote the discussion to make it as much easy to follow as possible for all readers. Could the Reviewer be more specific on how the discussion should be improved to be more clear for new readers?

Point 10: Add Comparison of the performance of the proposed model with the existing models for better understanding.

Response 10: Several performance comparisons between Leibovich 2018, GRANT and the novel nomogram have been reported throughout the manuscript. Specifically, in lines 209-224 we compared the calibration and DCA of the three models; in lines 294-306, we compared the accuracy, calibration and DCA of the novel nomogram with the ones of Leibovich 2018 and GRANT risk categories.

Point 11: Please revise all the grammatical errors in this manuscript carefully.

Response 11: We thank the reviewer for the suggestion. We checked the grammar of the whole manuscript carefully

Reviewer 4 Report

Mattia Luca Piccinelli and co-authors present an analysis of 3 models used to predict cancer-specific survival in 50 North American non-metastatic chRCC patients. They reported that all models give individual predictions that might be suboptimal. 

1. Figures and tables should be improved. The same font and font size are needed. The titles and text on the graphs are too small and very difficult to read. 

2. The authors clearly expressed that all studied models have limitations. However, other published papers presented these models' limitations and even proposed new ones. So, the scientific contribution of the work must be clearly stated. 

3. It would be advisable to add a clear proposal for improvement in the conclusions.

Author Response

Response to Reviewer 4 Comments

Mattia Luca Piccinelli and co-authors present an analysis of 3 models used to predict cancer-specific survival in 50 North American non-metastatic chRCC patients. They reported that all models give individual predictions that might be suboptimal.

We thank the reviewer for the comment. We hope that the following modifications will improve the quality of our work and satisfy the Reviewer.

Point 1: Figures and tables should be improved. The same font and font size are needed. The titles and text on the graphs are too small and very difficult to read.

Response 1: We thank the reviewer for the comment. We improved the figures' quality and enlarged the font.

Point 2: The authors clearly expressed that all studied models have limitations. However, other published papers presented these models' limitations and even proposed new ones. So, the scientific contribution of the work must be clearly stated.

Response 2: We thank the reviewer for the comment. The scientific contribution of the work can be summarized as follows:

  • The present represents the first validation of Leibovich 2018 and GRANT risk categories for the prediction of CSS in a North American chRCC population.
  • The only other external validation of Leibovich 2018 and GRANT risk categories for the prediction of CSS in chRCC relied on a cohort of 127 patients. Conversely, we provided the largest chRCC validation cohort analyzed to date (n=2,761).
  • Previous analyses included very historical patients in order to reach a sufficient number of events to develop or validate the models. Conversely, we relied on a relatively contemporary cohort (2000-2019).
  • We proposed a novel prognostic model based on the largest chRCC North American cohort analyzed to date.
  • Despite the large sample size and long follow-up, all models demonstrated important methodological limitations related to their performance and no model can be recommended above its competitors. Moreover, limitations related to the natural history of surgically treated non-metastatic chRCC (where few deaths occur relative to ccRCC) question the applicability and usefulness in the clinical practice of multivariable prognostic models. In consequence, the use of observational data that are stratified according to patient and tumor characteristics may, to date, prove more useful than the input of various prognostic models.

We modified the Introduction to better explain the scientific contribution of the work as follows:

Introduction, lines 67-74:

“Of these, two models are recommended by European guidelines to predict cancer control outcomes after nephrectomy in non-metastatic chRCC [11]: Leibovich 2018 [2,12] and GRade, Age, Nodes and Tumor (GRANT) [3,12,13]. However, their ability to predict cancer-specific survival (CSS) has been only validated in a mono-institutional European cohort of 127 chRCC patients [12]. We assessed the ability of these prognostic models to predict CSS in a population-based North American cohort relying on 2000-2019 Surveillance, Epidemiology, and End Results (SEER) database [14].”

We also modified the conclusion accordingly.

Point 3: It would be advisable to add a clear proposal for improvement in the conclusions.

Response 3: We thank the reviewer for the comment. We added the following sentence to the conclusion.

Conclusions, lines 336-338

“Further investigations regarding established and novel predictors of CSM and relying on large sample sizes with lon

Round 2

Reviewer 2 Report

the author addressed all the comments and the current version of the manuscript recommended for publication.

Reviewer 3 Report

It can be accepted now

Reviewer 4 Report

The authors addressed all comments. In my opinion, the manuscript is ready to be published.